# Strain-insensitive viscoelastic perovskite film for intrinsically stretchable neuromorphic vision-adaptive transistors

Chengyu Wang[1,2], Yangshuang Bian[1,2], Kai Liu[1,2], Mingcong Qin[1,2], Fan Zhang[1,2], Mingliang Zhu[1,2], Wenkang Shi[1,2], Mingchao Shao[1,2], Shengcong Shang[1,2], Jiaxin Hong[1,2], Zhiheng Zhu[1,2], Zhiyuan Zhao[1,2], Yunqi Liu [1,2] & Yunlong Guo [1,2] ✉

Stretchable neuromorphic optoelectronics present tantalizing opportunities for intelligent vision applications that necessitate high spatial resolution and multimodal interaction. Existing neuromorphic devices are either stretchable but not reconcilable with multifunctionality, or discrete but with low-end neurological function and limited flexibility. Herein, we propose a defect-tunable viscoelastic perovskite film that is assembled into strain-insensitive quasi-continuous microsphere morphologies for intrinsically stretchable neuromorphic vision-adaptive transistors. The resulting device achieves tri-chromatic photoadaptation and a rapid adaptive speed (<150 s) beyond human eyes (3 ~ 30 min) even under 100% mechanical strain. When acted as an artificial synapse, the device can operate at an ultra-low energy consumption (15 aJ) (far below the human brain of 1 ~ 10 fJ) with a high paired-pulse facilitation index of 270% (one of the best figures of merit in stretchable synaptic phototransistors). Furthermore, adaptive optical imaging is achieved by the strain-insensitive perovskite films, accelerating the implementation of next-generation neuromorphic vision systems.

Advancing neuromorphic optoelectronic systems to mimic visual perception, adaptation, and imaging are essential for next-generation artificial intelligence equipment, such as medical auxiliary gadgets, augmented reality displays and bionic robots[1-4]. All-in-one neuromorphic vision device that can substantially improve the computing efficiency and elegance, has provoked intensive interest in computer vision and deep learning[5,6]. To achieve such high-level optoelectronics, innovations in photosensing materials, compact structural design and integration technology are indispensable[7-10]. Extensive efforts have been devoted to endowing photosensors with synaptic or adaptive behavior through developing two-dimensional materials with tunable energy levels[11-13], engineering perovskite heterostructures[14-16] or building multilayer organic heterostructure[17,18]. However, the reported bioinspired optoelectronic devices and circuits generally exhibit monotonous functionality and severe performance degradation under mechanical deformations[19,20], which cannot satisfy the multiple demands imposed by wearable and implantable electronics.

Elastic neuromorphic vision sensors with excellent biocompatibility and mechanical compliance allow seamless attachment onto complex-shaped surfaces, naturally exhibiting high resolution and accuracy in mobile detection[21,22]. Recently, intrinsically stretchable neuromorphic optoelectronics have gained extensive attention owing to their free deformation and cross-scale modulus adaptability[23]. In particular, the intrinsically stretchable design undoubtedly enables more convenient fabrication, higher versatility and availability than complicated geometric engineering, becoming an inevitable componence for next-generation human-oriented applications. Notably, intrinsically stretchable organic phototransistors that have

[1]Beijing National Laboratory for Molecular Sciences, Key Laboratory of Organic Solids, Institute of Chemistry Chinese Academy of Sciences, 100190 Beijing, China. [2]School of Chemical Sciences, University of Chinese Academy of Sciences, 100049 Beijing, China. ✉e-mail: guoyunlong@iccas.ac.cn

demonstrated low noise and high photoconductive gain, are capable of functional integration of vision and synapses[19,20,24]. Despite recent progress in adaptive phototransistors[11–13,17,18,25,26], to date no case of the intrinsically stretchable adaptive sensor has been reported due to the stiff active materials and complicated architectures. Meanwhile, additionally introducing synaptic characteristics remains a daunting challenge for adaptive sensors with defect tunability, whether developing ionic conducting elastomer[27–29], assembling multiple monofunctional devices[30] or utilizing bilayer semiconductors[31]. Toward high-efficiency processing and low-power multiplexing, innovations in broadband vision-inspired neuromorphic sensors are extremely desirable[32]. However, existing stretchable photosensitive materials typically have a narrow absorption range or limited mechanical ductility that no longer pertain[20,24,33–35]. Although efforts have been devoted to combine perovskites with elastomer to balance stretchability and photoconversion performance[36], achieving strain-insensitive neuromorphic vision is still in its infancy. To tackle these challenges, exploiting excellent stretchable photosensitive materials and multifunction-integrated design is requisite for realizing intrinsically stretchable neuromorphic optoelectronic devices.

Herein, we report an intrinsically stretchable neuromorphic vision-adaptive transistor (ISNVaT) made of the defect-tunable viscoelastic perovskite film. The perovskite dots are microspherically distributed in elastomers by surface energy-induced strategy, enabling the strain-insensitive quasi-continuous microsphere (QCM) morphology. Such viscoelastic perovskite films not only ensure intrinsic stretchability and retentive photosensitivity, but also provide tunable charge-trapping defects that can guide photoadaptation and synaptic behaviors. Employing QCM perovskite film and elastic semiconductor layer to construct the layer heterojunction, the obtained device presented trichromatic photoadaptation and high biaxial stretchability (up to 100%). For synapse simulation, the ISNVaT showed an ultra-low energy consumption of 15 aJ and a record high paired-pulse facilitation (PPF) index of 270% (that is the highest value among reported stretchable synaptic phototransistors). Furthermore, a fast adaptive speed down to 150 s was realized, promising high-resolution adaptive imaging that was comparable with human eyes. ISNVaT technology pushes forward the skin-like neuromorphic vision systems for the emerging applications including visual prosthetics, bioinspired robots and unmanned intelligence.

## Results

### Defect-tunable perovskite films with QCM morphology

In retina, two photoreceptors (rod and cone cells) collaboratively realize polychromatic vision and dynamic adaptation in various light intensity, and then the visual information is processed and transmitted by the brain and visual cortex[12,18] (Fig. 1a). Neuromorphic optoelectronic device can integrate these functions into one device that largely improves the circuit reliability and processing efficiency of artificial vision systems[20,31,37]. Inspired by this, the elastic perovskite-based ISNVaT was developed to perform broad-wavelength detection, dynamic modulation and adapting imaging. Perovskite quantum dots (PQDs) with high photosensitivity and structural diversity are promising candidates for multifunctional photoelectronic devices[38,39]. However, the brittle PQD films have severe limitations on stabilizing the performance and integrating with stretchable materials. Herein, we proposed a universal fabrication process of viscoelastic photosensitive films composed of $CsPbBr_3$ quantum dots and SEBS elastomer (Fig. 1a and Supplementary Fig. 1) for stretchable neuromorphic vision devices. For a hybrid system, it is crucial to modulate the phase separation and self-assembly process, which could determine the ultimate film morphology and properties. The large gap of surface energies between SEBS (27.29 mJ m$^{-2}$) and PQDs (50.99 mJ m$^{-2}$) could induce uncontrollable phase separation behaviors, resulting in heterogeneous and rough hybrid films. To obtain the desired viscoelastic perovskite

film, we regulated the surface energy of the PDMS substrate from 9.42 to 65.72 mJ m$^{-2}$ (Supplementary Fig. 2, Supplementary Table 1), that could effectively adjust the surface morphology of hybrid perovskite films by controlling self-assembly behaviors (Supplementary Fig. 3, 4). Supplementary Fig. 5 shows a series of atomic force microscope (AFM) images of the hybrid films to reveal the detailed micromorphological evolution. At a low surface energy of the PDMS substrate, SEBS elastomer dominated the aggregation in the deposition self-assembly process, and a spindle-like hybrid film could be formed. With the surface energy increased, the PQDs tended to be dispersed, pullulating a honeycomb-like morphology and then a smooth quasi-continuous microsphere morphology. Compared with neat films, the optimized hybrid films with QCM morphology exhibited lower roughness, improved mechanical ductility and viscoelasticity (Fig. 1b, Supplementary Fig. 6, 7, and Supplementary Table 2). Notably, the SEBS elastomer processed high intrinsic viscoelasticity, so that the elastic modulus of hybrid perovskites films substantially reduced from 267 to 69.7 MPa (Supplementary Fig. 7). Additionally, the calculated adhesion work further demonstrated the optimized properties from neat films (60.13 mJ m$^{-2}$) to hybrid films (10:3, 67.99 mJ m$^{-2}$), thus proving that the optimized hybrid perovskites film enabled high viscoelastic property for the ideal interfacial contact[40] (Supplementary Table 2). Besides, it could be observed that the cubic crystalline structures and optical properties are mostly maintained with the introduction of SEBS elastomer and morphological transformation of PQDs, especially in the case of viscoelastic QCM perovskite films with a 10:1 ratio of $CsPbBr_3$ and SEBS (Fig. 1c–e and Supplementary Fig. 8, 9).

Photosensitive heterojunctions can promote the exciton dissociation efficiency and widen detection range for building robust and broadband photosensors[38,39]. Combining with the photoelectronic, and interface charge-trapping effect, we fabricated a fully elastic detect-tunable heterojunction consisting of a QCM hybrid film and a DPP-DTT/SEBS nanoconfined semiconductor film. The viscoelastic perovskite films with favorable quasi-continuous morphology have smooth surface and excellent strain-insensitive features, ensuring their reproducible transfer on the semiconductor layers (Supplementary Fig. 10). UV-vis absorption spectra and steady state photoluminescence were conducted to evaluate the optical properties of the elastic photosensitive heterojunction (Supplementary Fig. 9, 10). Figure 1f shows an absorption edge around 520 nm and a strong peak at 820 nm that corresponds to the characteristic peaks of PQDs and DPP-DTT semiconductor, respectively, verifying the non-destructive integration of the viscoelastic QCM hybrid film and semiconductor film. In addition, the obtained heterojunction exhibited obvious complementary absorption from near-infrared to visible, and to ultraviolet regions, guaranteeing the visual detection in a wide range of wavelengths. Benefiting from the compact stack and homology effect[40–42], the elastic heterojunction owned ideal interface for relatively high-effective charge transport even under 100% strain or released state (Supplementary Table 2, Supplementary Fig. 10). We also measured the transient photoluminescence decay spectra that demonstrated an evolved decay lifetime achieved by modulating the surface morphology of hybrid perovskite films[43,44], further proving the tunable defect states in the elastic heterojunction (Fig. 1g). Additionally, the hole-only devices measurement was applied to quantify the defect state density ($N_t$) of different perovskite films[45]). The defect states were certainly introduced from neat films (5.62 × 10$^{18}$ cm$^{-3}$) to higher elastomer content films (10:3, 1.08 × 10$^{19}$ cm$^{-3}$), and could be effectively tuned through surface energy-induced strategy form spindle-like morphology (7.96 × 10$^{18}$ cm$^{-3}$) to QCM morphology (5.79 × 10$^{18}$ cm$^{-3}$) (Supplementary Fig. 11 and Supplementary Table 3).

### Realization of visual adaptation by ISNVaTs

With excellent stretchability and defect state tunability, our elastic photosensitive heterojunction becomes a cornerstone for intrinsically

stretchable neuromorphic vision devices. We applied such hetero-junctions into intrinsically stretchable organic transistors with a bottom-gate-top-contact configuration to simultaneously emulate visual and neuromorphic functions (Fig. 2a). Through sequential thermal lamination and transfer procedures (Supplementary Fig. 12), the defect-tunable heterojunction and ideal contact interface between dielectric and semiconductor layer were achieved that collectively ensured the multiple optical neurobehaviors of ISNVaTs. We investigated the optoelectronic properties of hybrid perovskite films with different SEBS contents and surface morphology. The optimum performance including a remarkable light intensity-dependent modulation, a high photocurrent/dark current ratio ($I_{light}/I_{dark}$) up to $10^5$ and excellent operational stability were observed in the ISNVaTs based on elastic QCM films for ultraviolet illumination (Fig. 2b–d and Supplementary Fig. 13–16). Noted that the QCM morphology provided ideal heterocontact and sufficiently effective transport of photo-induced

carriers, resulting in stable photogating behaviors (Supplementary Fig. 17, 18). Along with the high photoresponse to ultraviolet light, the prepared ISNVaTs also showed obvious photoswitching behaviors under visible, and near infrared light, which was mainly ascribed to the wide absorption of elastic heterojunction (Supplementary Fig. 19). To perform the comparative experiments of neat and hybrid PQD films, we measured the temporal photoresponse curves of ISNVaTs, in which the defect-tunable QCM film showed a slow recovery of photocurrent (Supplementary Fig. 20). We then studied the relationship of the photosensitivity ($S_{ph}$) and lighting power intensity ($P_{in}$) under different gate biases. The $S_{ph}$ gradually increased with intensifying $P_{in}$, and presented a positive correlation with the gate bias (Supplementary Fig. 21). The photoresponse characteristics of three successive illumination pulses with variable intensities or under different gate bias were shown in Fig. 2e and Supplementary Fig. 22. The normalized $I$-$T$ curves intuitively revealed the critical effects of $P_{in}$ and $V_G$, in photodetection,

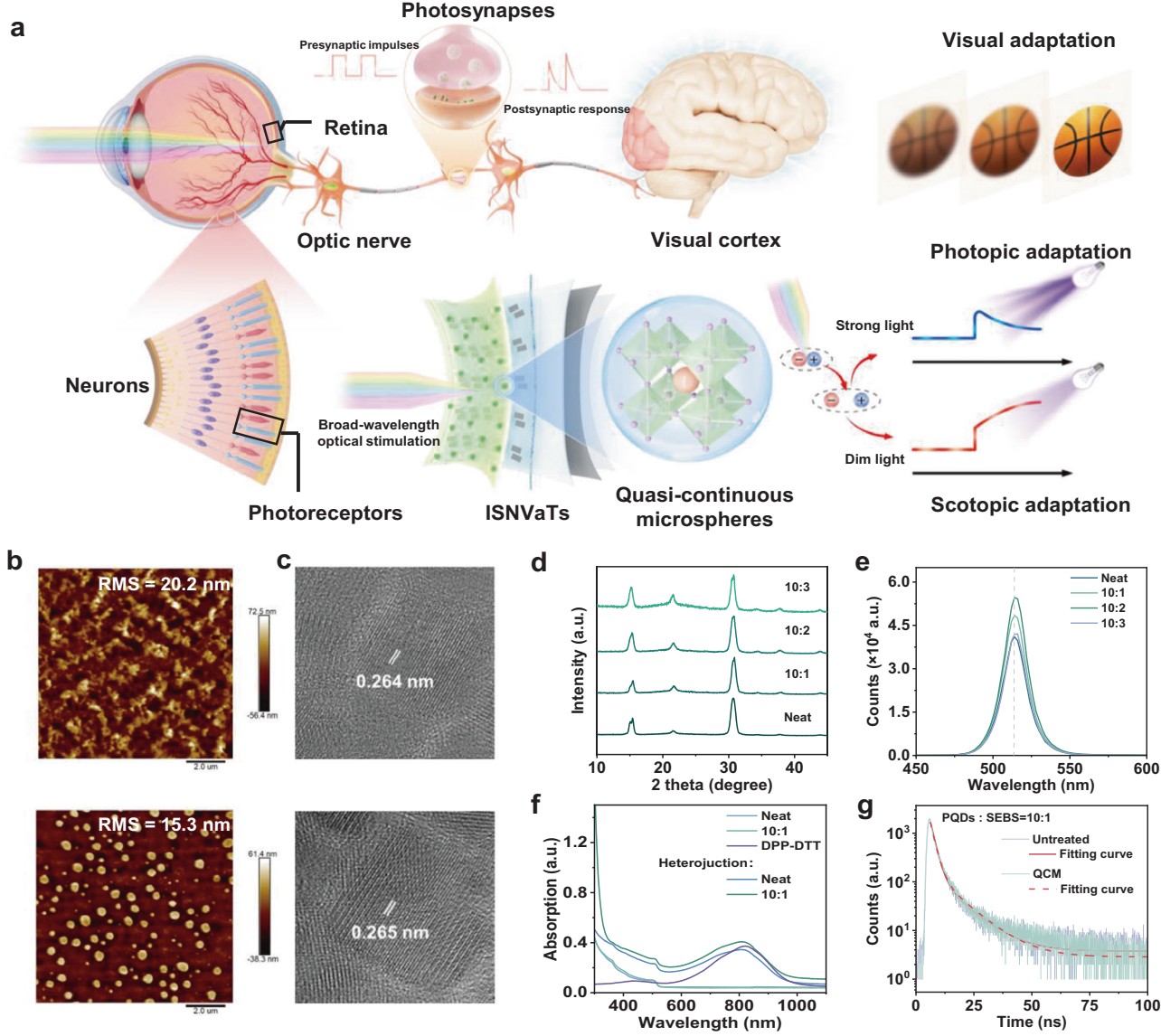

**Fig. 1 | Strain-insensitive viscoelastic perovskite films for neuromorphic visual adaptation. a** Visual adaptation mechanism of human eyes. Photoreceptors (including cones and rods) and horizontal cells together contribute to accurate representation under extreme light conditions. **b** Atomic force microscope (AFM) images of the neat and hybrid perovskite films with optimum QCM morphology at a ratio of 10:1 perovskite and SEBS with $O_2$ plasma treatment on the substrate of 6 min. **c** Transmission electron microscope (TEM) images of the neat and optimized

hybrid perovskite films. **d** X-ray diffraction patterns of the neat and hybrid perovskite films with different SEBS contents. **e** Steady state photoluminescence (PL) of the neat and hybrid perovskite films with different SEBS contents. **f** UV-vis absorption spectra of different photosensitive films including elastic DPP-DTT film, neat and hybrid perovskite film, and their corresponding layer heterojunctions. **g** Transient PL decay spectra of the initial and optimized hybrid photosensitive films onto elastic layer heterojunctions.

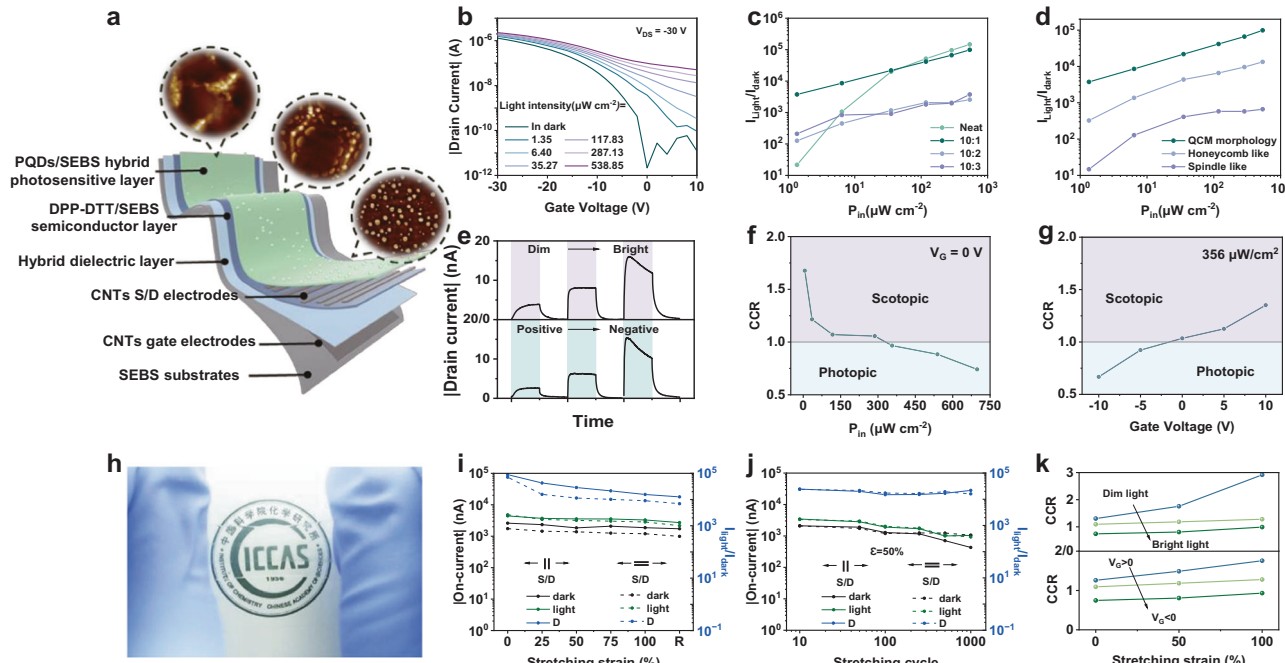

**Fig. 2 | Vision-adaptive behaviors of ISNVaTs. a** Schematic of the intrinsically stretchable neuromorphic vision-adaptive transistor (ISNVaT) based on elastic heterojunction. **b** Typical transfer curves of the ISNVaT with a viscoelastic QCM perovskite film under ultraviolet illumination with various intensities. **c** Switching current ratio ($I_{light}/I_{dark}$) as a function of $P_{in}$ extracted from ISNVaTs based on neat and hybrid perovskite films with various SEBS contents. **d** Switching current ratio ($I_{light}/I_{dark}$) as a function of $P_{in}$ extracted from ISNVaTs based on hybrid perovskite films (PQD: SEBS = 10:1 w/w) with different surface morphologies. **e** Light intensity dependence (top, $V_G = 0$) and gate voltage dependence (bottom, $P_{in} = 356.69 \ \mu W \ cm^{-2}$) adaptation behaviors of ISNVaTs. adaptation behaviors of ISNVaTs. (dim: 6.41 $\mu W \ cm^{-2}$, normal: 356.69 $\mu W \ cm^{-2}$, bright: 698.09 $\mu W \ cm^{-2}$; positive: +10 V, normal: 0 V negative: −10V). **f** Current change ratio (CCR) extracted at different lighting power density ($P_{in}$) and **g** different gate voltages ($V_G$) **h** Photograph of an array of ISNVaTs under the mechanical strain. **i** Strain-tolerance characteristics of ISNVaTs. **j** Tensile stability of ISNVaTs with different stretching cycles. **k** CCR extracted at different stretching strains demonstrating the trends of light-intensity-dependence (top) and gate-voltage-dependence (bottom) under ultraviolet light.

where the current exhibited obvious tunable growth or decay. These results indicated that light intensity and gate voltage together modulated the photoelectric behaviors of ISNVaTs, enabling the potential to emulate negative feedback functions in the visual system.

As an important measure used to characterize the adaptive performance of ISNVaTs, the current change ratio (CCR) is defined according to the rod-cone break in biological system as follows[12]:

$$CCR = \frac{I_{D-50s}}{I_{D-10s}} \tag{1}$$

where $I_{D-10s}$ and $I_{D-50s}$ represent the source-drain current under light illumination for 10 s and 50 s, respectively. (Supplementary Fig. 23) The CCR of common phototransistors is generally around 1 with a relatively stable photocurrent under different light intensities. To benchmark adaptive performance, CCR > 1 generally is denoted as the photocurrent excitation, and <1 indicates the inhibition. For our ISNVaTs, the calculated CCR had a strong dependence on the $P_{in}$ at gate voltage ($V_G$) of 0 V, displaying the obvious excitation and inhibition effects of the light intensity on the drain-source current ($I_{DS}$) (Fig. 2f and Supplementary Fig. 24). Then we further evaluated the photo-adaptive characteristics of ISNVaTs by applying various $V_G$ at a fixed illumination condition of 356 $\mu W \ cm^{-2}$. The excitation and inhibition behaviors were also observed when applying positive and negative $V_G$, respectively (Fig. 2g and Supplementary Fig. 25). The scotopic adaptation was achieved under a dim light of 6.41 $\mu W \ cm^{-2}$ with the highest CCR value of 1.68, whereas the photopic adaptation under a bright background of 698.09 $\mu W \ cm^{-2}$ with the lowest CCR value of 0.74 at $V_G = 0$ V (Fig. 2f, g and Supplementary Fig. 24, 25). Therefore, the ISNVaTs could simulate both scotopic and photopic adaptation of visual adaption processes by time-varying excitative and inhibitive photocurrent, in which the light intensity and gate voltage were critical to modulate the adaptive behaviors and processed reliable statistical validation (Supplementary Fig. 26). It was worth mentioning that all the photoadaptation could be achieved by ISNVaTs in the detectable wavelengths, particularly for the visible light region. With the increase of wavelength, the CCR curves showed an obvious drift to the high illumination intensity, displaying a better scotopic adaptation than ultraviolet light (Supplementary Fig. 27a). In the gate-voltage-dependent characteristics, similar excitation and inhibition behaviors were observed despite the low utilization of visible light that resulted in a weak dominance of $V_G$ (Supplementary Fig. 27b).

Besides remarkable simulation and tunability of visual adaptation, high mechanical durability is required for skin-like vision systems to sufficiently cope with various buckling movements of human body involving strains from 20 to 80%. As shown in Fig. 2h, our ISNVaTs presented high transparency and stretchability, demonstrating their potential usage in wearable and implanted electronics. Since the devices were made in fully stretchable materials, their photoelectrical properties behaved relatively stable under mechanical strains. For example, the on-current ($I_{on}$) and photocurrent/dark current ratio ($I_{light}/I_{dark}$) did not occur significant degradation when the ISNVaTs were subjected to stretching strains up to 100% in the directions parallel or perpendicular to the carrier transport (Fig. 2i, Supplementary Fig. 28). Moreover, the photoswitching behaviors of the devices could be maintained even after 1000 cycles of stretching-releasing test up to 50% strain along the direction parallel to the channel (Fig. 2j, Supplementary Fig. 29). These excellent mechanical properties were achieved by allowing the strain dissipation in the QCM photosensitive films to be mainly localized on the elastomer, thereby minimizing the deformation of the perovskite microspheres. Significantly, benefiting from the strain insensitive QCM photosensitive films, our ISNVaTs remained

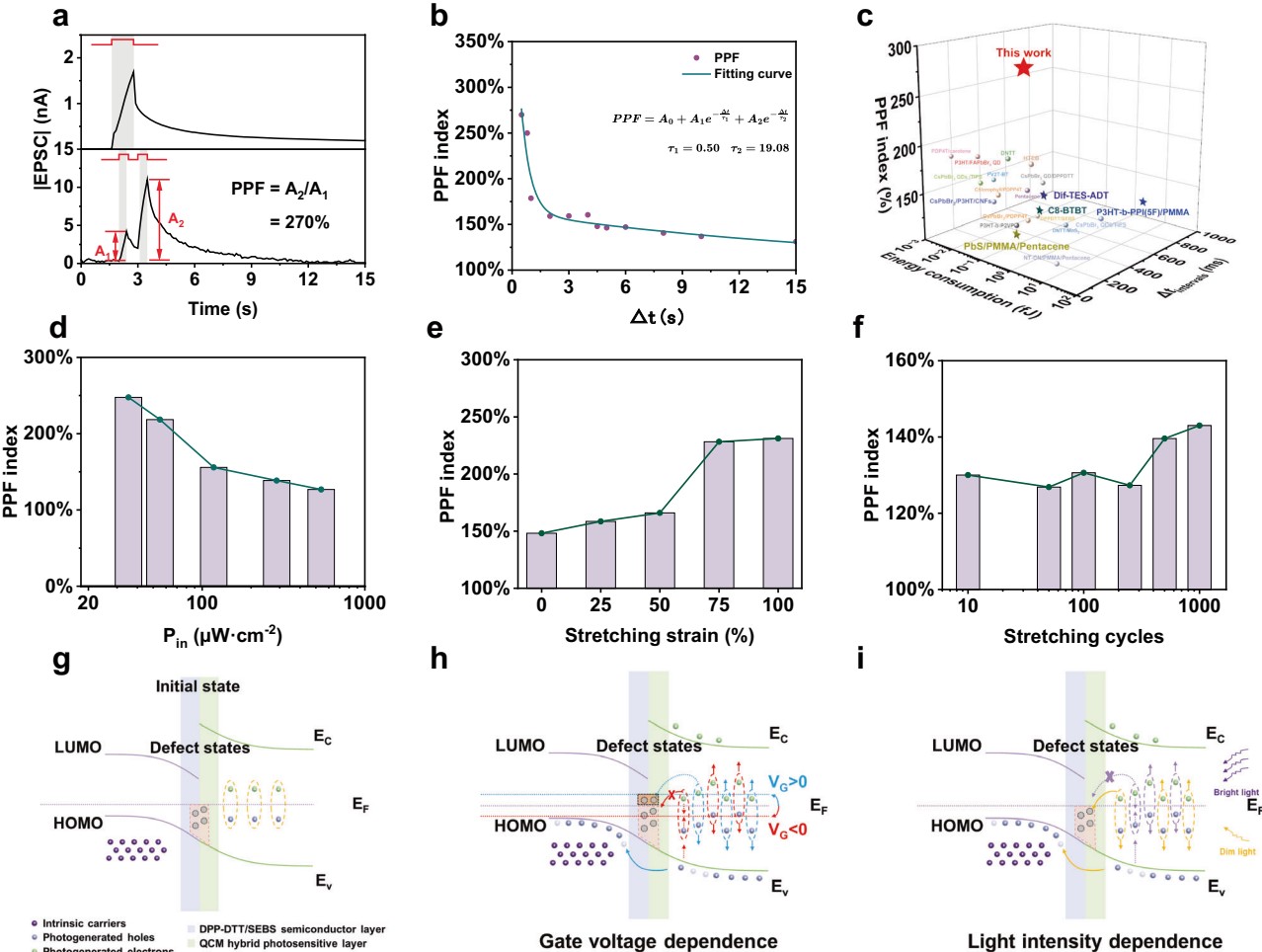

**Fig. 3 | Synaptic behaviors of ISNVaTs. a** Typical single pulse enhancement and PPF behaviors of ISNVaTs with the $P_{in}$ of 6.41 μW cm$^{-2}$ and 35.27 μW cm$^{-2}$, respectively. **b** PPF index at different light pulse intervals with the same light pulse pair (35.27 μW cm$^{-2}$, 0.5 s). **c** Comparison of the PPF index and energy consumption achieved by previously reported synaptic organic phototransistors with our work. **d** PPF index at different light intensities with the illumination time and $\Delta t$ of 0.5 s. **e** PPF index under different stretching strains and **f** after different stretching cycles under the same pair illumination condition (356.69 μW cm$^{-2}$, 0.5 s) and the $\Delta t$ of 0.5 s. **g** Schematic illustrations of the initial state energy-band diagrams and mechanism processes of **h** gate-voltage-dependence adaptation and **i** light-intensity-dependence adaptation of ISNVaTs.

excellent scotopic adaptation behaviors even under strain, only with a slight influence on the photopic adaptation (Fig. 2k, Supplementary Fig. 30). The results demonstrated the enhanced scotopic adaptation of the devices with a high CCR value of 2.92 on stretching by 100% that was twice than the initial state. That could be attributed to the enlarged active region and increased defect states on the charge transport interface under strains, which extended electron-trapping duration and thus facilitated photocurrent excitation processes. The desirable strain-insensitive photoelectric properties and stable visual adaptation behaviors under strains make the ISNVaTs promising candidates for intelligent visualization applications.

## Simulation of photonic synapses for ISNVaTs

Synapses are essential tools and bridges for transmitting neural signals in visual system[46,47], and the synaptic plasticity holds the key to brain memorization that can be classified into short-term plasticity (STP) and long-term plasticity (LTP) according to the retention time[13,48]. To date, the reported synaptic devices were manufactured with rigid components or ionic format materials that precluded the stretchable integration of visual adaptation and synaptic functions[12,18,47,49]. In this work, the developed ISNVaTs with time-dependent response and memory characteristics could easily merge the phoroadaptation and photosynaptic behaviors even without the involvement of unique architecture design.

Firstly, we performed the basic synaptic characteristics by applying various optical pulses. As depicted in the top panel of Fig. 3a, the device demonstrated typical excitatory postsynaptic current (EPSC) response when single optical pulse was applied to the device. We attributed this to the increased current in the channel and the long-decayed photo-generated electrons trapped inside the elastic perovskite film[50]. Notably, an ultra-low consumption of 15 aJ was achieved in our ISNVaTs by applying single optical pulse with a $P_{in}$ of 6.41 μW cm$^{-2}$ at $V_{DS}$ = −1 μV, which was record low among the reported flexible or stretchable organic photosynaptic transistors (Supplementary Fig. 31, 32 and Supplementary Table 4). As a figure of merit for quantifying the STP, paired-pulse facilitation (PPF) describes the enhancement effect of EPSC by two successively applied light interspikes (Fig. 3a, bottom panel). The PPF index can be defined as followed[7]:

$$\text{PPF} = \frac{A_2}{A_1} \times 100\% \qquad (2)$$

where $A_1$ and $A_2$ are the peak amplitudes of the EPSC induced by the first and second optical simulations, respectively. The calculated PPF index degraded gradually when increasing the interspike intervals ($\Delta t$) due to the decrease in trapped electrons, as a result of the reduced internal electric field (Fig. 3b). Therefore, the device manifested a high PPF index

up to 270% under a light pulse pair (35.27 µW cm$^{-2}$, 0.5 s) with interspike intervals ($\Delta t$) of 0.5 s by applying $V_G$ and $V_{DS}$ of 0 V and −10 V, respectively (Fig. 3a, b). In contrast to previously reported flexible synaptic phototransistors based on organic semiconductors[51–70], our ISNVaTs simultaneously achieved an ultra-low energy consumption and a record high PPF index (Fig. 3c, Supplementary Fig. 33, Supplementary Table 5, 6). Furthermore, the PPF index was also strongly dependent on the light intensities and showed a similar negative correlation with the $\Delta t$ (Fig. 3b, d). Fitting the PPF index curves by a double exponential decay function, the fitting results indicated a rapid relaxation time ($\tau_1$) of 0.50 and a slow relaxation time of 19.08 that were well corresponding to the biological synapses while $\tau_2$ was one order of magnitude larger than $\tau_1$[13] (Fig. 3b). To emulate the learning and memory pattern of human brain, we then demonstrated the transition from the STP to the LTP in ISNVaTs by altering light pulse duration (Supplementary Fig. 34). Notably, our ISNVaTs exhibited unique strain-enhanced synaptic behaviors when subjected to 100% strain or 1000 cyclic stretching because of the strain-induced defect states that led to the unsaturated photogenerated charge injection and the strengthened memory retention[20,31,37] (Fig. 3e, f).

It has been verified that the relaxation of trapped charges and tunable interface defects were responsible for the synaptic[13,50] and vision-adaptive[12,18] characteristics, respectively. In details, the photogenerated excitons were rapidly separated under illumination, while the holes were injected into the conductive channel and the electrons were trapped by the defect states, allowing the photocurrent to increase slowly. The synaptic plasticity of ISNVaTs was achieved by the long decay time of trapped electrons as mentioned previously. To further elucidate the mechanism of adaptive behaviors in ISNVaTs, we briefly described the possible charge transport and defect state tunability during the adaptation processes (Fig. 3g–i, Supplementary Fig. 35–38). Different from the traditional photodetectors with a constant saturated photocurrent, our devices achieved the relative position shift between the interface states and the quasi-Fermi level ($E_F$) by designing defect-tunable elastic heterojunctions under various illumination conditions, thus achieving dynamic balance and self-modulation of photocurrent. As depicted in Fig. 3g, the photogenerated holes typically injected into the HOMO level of the elastic semiconductors, whereas the electrons were trapped into the defect states in the initial state (Supplementary Fig. 35). When subjected to positive (negative) $V_G$, the $E_F$ could be raised (lowered) under higher bias, playing a leading role on releasing trapped electrons into the conductive channel to promote (weaken) the recombination with holes (Fig. 3h, Supplementary Fig. 36). With the increase of illumination intensity, the photogenerated excitons were liberally produced, in which the photogenerated electrons gradually filled the defect states or recombined with photogenerated holes resulting in decreased current. However, the recombination and charge transport were limited by the intrinsic trap states and the heterojunction effect which would lead to generally saturated current under constant illuminations (Fig. 3i, Supplementary Fig. 37). Moreover, the effect of wavelength on adaptive behavior was also addressed (Supplementary Fig. 38). Therefore, the ISNVaTs enabled by coupling adaptive modulations of gate voltage and light intensity, achieved excellent adaptive performance to light irradiation, including current excitation effect at dim light and current inhibition effect under strong illumination background.

### Vision-adaptive imaging with an ISNVaT array

Given the excellent adaptation capabilities of the ISNVaTs, we successfully emulated the vision-adaptive functions of human eyes, including the active processes of natural photopic and scotopic adaptation (Fig. 4a). To demonstrate spatially resolved images, a 5 × 5 transistor array with the small device-to-device variation was constructed to mimic the visual adaptation behaviors. For an intuitive description, we designed a special distribution in the 'X' pattern, where the central pixel was exposed to an appropriate light background as a reference, and the peripheral 8 pixels were subjected to extreme illumination intensity (Fig. 4b, c). Converging pixel current to the same level, the normalized images could be obtained under different illumination periods. By virtue of time-varying characteristics in the ISNVaTs, high-contrast X-shaped images were provoked in the instantaneous exposure of dim light (Fig. 4b). However, the corresponding distribution became blurry during the continue illumination of bright light environment (Fig. 4c). Notably, even under 50% stretching stain, the ISNVaT array was able to display undiminished vision-adaptive functions and clear images under extreme illumination. Additionally, the ISNVaT array also demonstrates excellent gate voltage dependence with high-contrast images at positive gate voltages and slight chromatic aberration images at negative gate voltages (Supplementary Fig. 39). These results implied that our ISNVaT array exhibited the remarkable photopic and scotopic adaptation with a faster response (< 150 s) than the human eyes (3–30 min), that was also comparable to existed devices (Supplementary Table 7). More importantly, adaptive imaging was implemented for the first time by intrinsically stretchable phototransistors, suggesting a hopeful outlook for elastic visual prosthetics.

As a benefit of the steady outputs under different light intensities, the fabricated neuromorphic vision devices showed extensive prospect for next-generation intelligent imaging. To discriminate the complex objects, we also performed the digital images using a single-pixel testing system with a stepper platform (X–Y direction) and a readout circuit that were all controlled by a computer. Time courses of adaptive imaging were obtained by moving a basketball-shaped mask with an intermittent sampling mode of 10, 100, 1000 ms, respectively (Fig. 4d). As the interval increased, a clear image of the basketball gradually emerged with an increasingly high image contrast because of the dynamically enhanced photocurrent under the extended light duration. The reliability and generalizability of the single-pixel adaptive imaging were also certified by different patterns and light stimulation (Supplementary Fig. 40). To further verify the potential of ISNVaTs in practical applications, we also demonstrated their strain-insensitive photoelectronic performance under dynamic stretching (Supplementary Movie 1 and 2). Furthermore, the light-intensity-dependent current excitation and inhibition phenomenon was revealed using an ISNVaT pixel that attached to the ocular prosthesis as a light-emitting diode (LED) driver (Fig. 5a, b), where the LED displayed the expected scotopic and photopic behaviors even under different wavelengths (Fig. 5c, d and Supplementary Movie 3, 4, 5). Consequently, we concluded that our devices, capable of in-sensor adaptation, were expected to reproduce the visual perception-adaptation-imaging loop for advanced intelligent neuromorphic electronics. In particular, ISNVaTs that can be seamlessly attached on the moving objects with excellent biocompatibility make them tantalizing for next-generation skin-like artificial intelligence equipment.

## Discussion

Through excellent designs in the viscoelastic perovskite films and device engineering, we achieved a trichromatic neuromorphic vision-adaptive sensor based on intrinsically stretchable phototransistors. The described viscoelastic perovskite film with the quasi-continuous microsphere (QCM) morphology features intrinsic stretchability, retentive photosensitivity and defect tunability that can guide photoadaptation and synaptic behaviors. The resultant phototransistors exhibit an ultra-low energy consumption down to 15 aJ, a record high paired-pulse facilitation (PPF) index of 270% and a high biaxial stretchability up to 100%. Furthermore, a fast adaptation speed (< 150 s) was realized which was able to realize adaptive imaging beyond human eyes (3–30 min). Therefore, our ISNVaTs pave the way for visual prosthetics, bioinspired robots and unmanned intelligence.

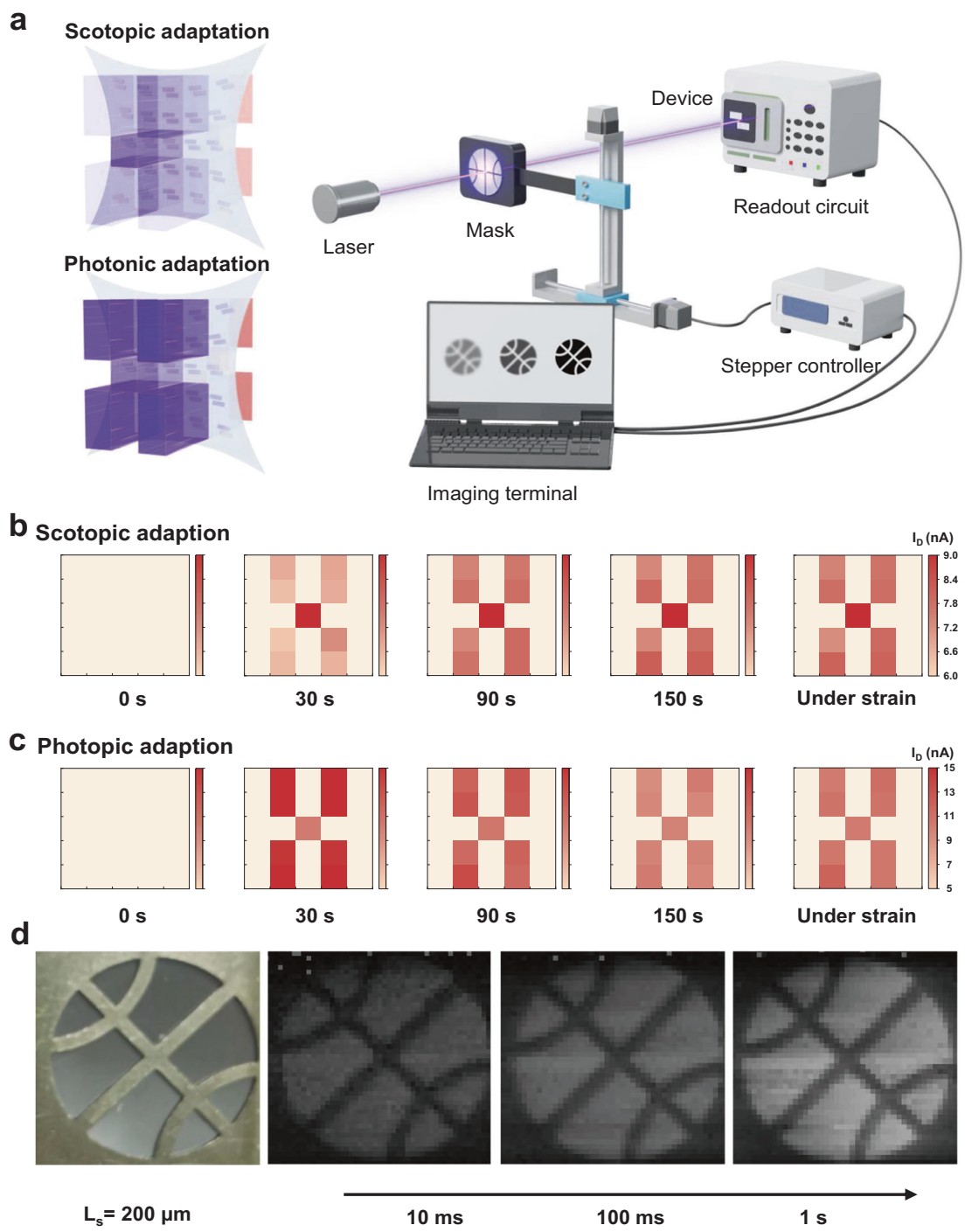

**Fig. 4 | Visual adaptation imagination. a** Schematic of an ISNVaT array and the procedures of single-pixel imaging. **b** Resulting current mapping of the 5*5 ISNVaT array at different time (0, 30, 90, 150 s and 150 s under 50% stretching strain) and under different conditions (dim: 6.41 μW cm⁻², normal: 356.69 μW cm⁻², bright: 698.09 μW cm⁻²) to simulate scotopic and **c** photopic adaptation. **d** Simulated visual adaptation results of ISNVaTs through imaging a basketball-shaped mask (illumination intervals: 10 ms, 100 ms and 1 s).

## Methods

### Materials and chemicals

All the processing solvents, such as chlorobenzene, toluene, cyclo-hexane, 2-propanol, were easily purchased from commercial sources and used as received. CsPbBr₃ perovskite quantum dots (CPB PQDs) (10 mg mL⁻¹, n-hexane) were purchased from Nanjing MKNANO Tech. Co. Ltd., with a PL peak of 513 nm and a full width at half-maximum of about 18 nm, using oleic acid (molecular formula: C₁₈H₃₄O₂) and oil ammonia (molecular formula: C₁₈H₃₇N) to be as capping agents.

The polymer (poly(3,6-di(2-thien-5-yl)−2,5-di(2-octyldodecyl)-pyr-rolo[3,4-c]pyrrole-1,4-dione)thieno[3,2-b]thiophene) (DPP-DTT) was supplied by Derthon Optoelectronic Materials Science Technology Co. LTD. (Shenzhen, China). Carbon quantum dots (CQDs) (10 mg mL⁻¹, toluene) with an average diameter of 4 nm, were supplied by Janus New-Materials Co., Ltd., and were covered with plenty of carboxylic groups and hydroxyl groups. SEBS (H1221 and H1052) was provided by Asahi Kasei company. SEBS H1052 with 80% volume fraction of poly(ethylene-co-butylene) was incorporated with CQDs as the

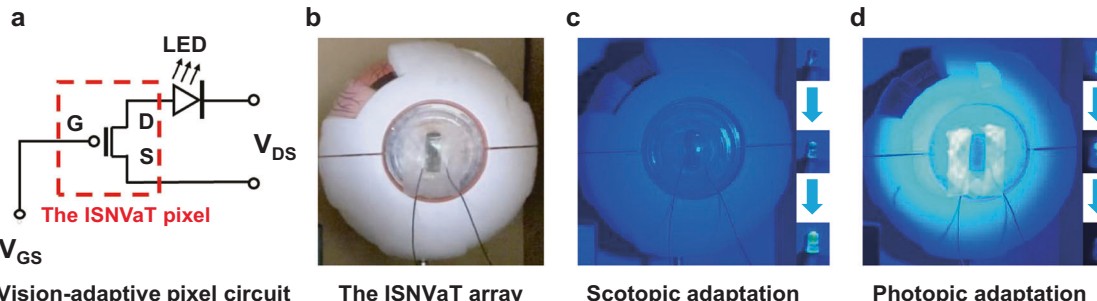

**a** Vision-adaptive pixel circuit    **b** The ISNVaT array    **c** Scotopic adaptation    **d** Photopic adaptation

**Fig. 5 | Visual adaptation of an ISNVaT pixel serving as an LED driver. a** Vision-adaptive pixel circuit schematic of the ISNVaT pixel for driving an LED. **b** The actual picture of the ISNVaT pixel was attached to an ocular prosthesis. **c** Scotopic adaptation of the ISNVaT pixel that drove the LED from dark to bright. **d** Photopic adaptation of the ISNVaT pixel that drove the LED from bright to dark.

dielectric layer in the stretchable devices. SEBS H1221 with 88% volume fraction of poly(ethylene-co-butylene) was employed as the stretchable substrate and encapsulation layer in the stretchable devices, and as the elastic component in the hybrid polymer semiconductor and photosensitive layer. Poly(dimethylsiloxane) (PDMS, Sylgard 184, base and crosslinker) was purchased from Dow Corning. Carbon nanotubes (P3-SWNTs) for gate, source and drain electrodes were purchased from Carbon Solutions.

### Preparation of hybrid photosensitive films

The SEBS-H1221 was directly added to the purchased $CsPbBr_3$ QDs solution (10 mg mL$^{-1}$, n-hexane) with different weight ratios (SEBS-H1221: $CsPbBr_3$ QDs) of 10:1, 10:2 and 10:3, followed by continuous stirring at room temperature (around 25 °C) for 1 h. The excellent photovoltaic conversion efficiency, matched energy levels, and good compatibility with elastomers collectively enabled the $CsPbBr_3$ perovskite QDs quite desirable for use in photosensitive layers. To tune the morphology and microphase separation of hybrid films, a high-qualitied surface-hydroxylation PDMS (SHPDMS) substrate was essential, which was fabricated by spin-coating PDMS (base/cross-linker = 10:1, w/w) solution (diluted in cyclohexane for 0.5 mg mL$^{-1}$) onto the Si wafer. Then the surface energy of SHPDMS was modulated by controlling the treatment times of $O_2$ plasma on the surface of PDMS substrates. After that, the hybrid PQD solution was spin-coated on the optimized PDMS substrate at 1500 rpm for 40 s without annealing treatment. As a result, the QCM perovskite films were successfully fabricated on the substrate with $O_2$ plasma treatment around 6 min (30 W, 0.5 air pressure).

### Preparation of constituent thin films

To optimize the transfer characteristics of various constituent layers, Si wafer was cleaned with the piranha solution (70 vol% $H_2SO_4$ + 30 vol% $H_2O_2$), deionized water, alcohol and acetone, and then modified by octadecyltrimethoxysilane (OTS) molecules to increase surface hydrophobicity. The CNT solutions (0.6 mg mL$^{-1}$) were spray-coated onto the OTS-modified Si wafer with designed mask for the gate and source/drain electrodes ($L/W = 200/4000$ μm). For hybrid semiconductor films, the polymer solutions (10 mg mL$^{-1}$) prepared by dissolving the conjugated polymers and SEBS with a weight ratio of 5:5 in chlorobenzene, were spin-coated onto the substrate at 3000 rpm, and then annealed at 150 °C for 20 min. For the dielectric layer, the SEBS-H1052 and carbon quantum dots (CQDs) were simultaneously dissolved in the toluene solvent at a specific weight ratio of 100:3.5. The dielectric film with a thickness of 2 μm was obtained by spin-coating the CQD/SEBS solution on the substrate at 1500 rpm, and then annealed at 90 °C for 40 min. Both of semiconductor and dielectric layers were prefabricated on OTS-modified Si wafers for the subsequent use. The SEBS-H1221 solution with 200 mg mL$^{-1}$ was dropped onto OTS-modified glass as a substrate.

### Fabrication of ISNVaTs

The ISNVaTs with a bottom-gate-top-contact (BGTC) configuration were fabricated through consecutive thermal lamination-transfer procedures. The elastomer substrate was employed to successively transfer the patterned gate electrodes, dielectric layer, nanoconfined semiconductor film, patterned source/drain electrodes from OTS-treated Si wafer, prefabricated hybrid photosensitive films from SHPDMS substrate, and the ultimate encapsulation layer from OTS-treated Si wafer. The measured thickness of each layer was about 30 nm, 50 nm, 50 nm, 1800 nm, 80 nm and 2 mm for the viscoelastic perovskite photosensitive layer, CNT source/drain electrodes, hybrid semiconductor layer, hybrid dielectric layer, CNTs gate electrode, and SEBS substrate, respectively. Each thermal lamination process was performed in a vacuum drying oven (about −0.1 MPa) at 60 °C for 15 min, and each transfer process was conducted at room temperature. Notably, the yield of device construction was around 92.3% obtained from 5 batches of 65 devices.

### Electrical and optical characterization

All electrical characteristics were measured using a Keithley 4200-SCS and a Keysight B2900A in a nitrogen-filled glovebox. UV-vis absorption spectra for photosensitive films, polymer semiconductor films, and layer heterojunctions films, were measured by the UV visible spectrophotometer (UH 4150, Hitachi). Steady-state PL spectra were obtained by fluorescence spectrophotometer (FLS980, Edinburgh). Time-resolved PL decay spectra were recorded and analyzed by a modular fluorescence and phosphorescence spectrometer (FLS1000, Edinburgh) with a pulse laser excitation source. For resultant hybrid photosensitive films, the optical images were collected by an Olympus BX51 cross-polarized optical microscope, the SEM images were taken with a Hitachi S-4800 field emission scanning electron microscope, the AFM images were obtained with NanoMan VS atomic force microscopy in the tapping mode, and the X-ray diffraction patterns were recorded by a Bruker D8 Advance XRD diffractometer with Cu Kα radiation. TEM and HRTEM images of the colloidal CPB QDs were captured by JEOL JEM-2100F TEM at 200 kV. The water ($H_2O$)- and diiodomethane ($CH_2I_2$)-contact angles of SHPDMS substrates were acquired by a motorized drop-shape analysis system (DSA100) from KRÜSS according to the Owens-Wendt method. Furthermore, for the photoelectrical characterization, multiple light sources with wavelengths of 365, 460, 520, 625 and 808 nm, were used to illuminate the active channel of the ISNVaTs. Photoswitching characteristics of the devices were investigated by modulating the incident illumination intensities that were determined by a calibrated Si-photodiode (818-UV, Newport).

## Data availability

All data that support the findings of this study are available within the article and its Supplementary Information or from the corresponding authors upon request. Source data are provided with this paper.

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

## Acknowledgements

This work was financially supported by the National Key R&D Program of China (2021YFB3200701), the National Natural Science Foundation of China (61890940, U22A6002 and 82151305), the CAS Project for Young Scientists in Basic Research (YSBR-053), the CAS-Croucher Scheme for Joint Laboratories, Lu Jiaxi international team (GJTD-2020-02), the CAS Cooperation Project (121111KYSB20200036), and the Beijing Nova Program (20220484173).

## Author contributions

C.W., Y.B. and K.L. contributed equally to this work. Y.G. and Y.L. proposed and supervised the project. Y.G., C.W., Y.B. and K.L. conceived the idea. C.W. designed the experiments. M.Q., F.Z. and W.S. conducted the photodetection experiments. M.Z., M.S., S.S. and J.H. were involved in the film characterizations. Z.Zhu and Z.Zhao were involved in the AFM characterizations. Y.B. conducted the imaging experiments. Y.G., C.W., Y.B. and K.L. wrote the manuscript and all authors reviewed it.

## Competing interests

The authors declare no competing interests.
