## [Peer Review File · Nature Communications]

REVIEWER COMMENTS

Reviewer #1 (Remarks to the Author):

In this manuscript, the authors used a defect-tunable viscoelastic perovskite film to fabricate intrinsically stretchable neuromorphic vision-adaptive transistors. The reviewer believes that researchers in this field will be interested in the contents of this manuscript. However, the mechanism part is not very clear to the reader. Thus, the reviewer suggests that this manuscript is suitable for publication in Nature Communication after minor revision. To strengthen the manuscript, several points seemed to be discussed in detail.

1. In this paper, the author used the transfer method to fabricate the devices. What is the yield? The reviewer recommended adding some statistical data.
2. What are the thickness for each layer?
3. There is only 0% and 100% result for OM data with stretchability. To show the stretchability more obviously, the reviewer recommended adding the intermediate state, like 20%, 30%, 50%, 70%, etc...
4. The method that perovskite materials combine with SEBS to improve the stretchability has been reported by other paper, Adv. Mater. 2020, 32, 2001989. Therefore, this method is not new. The reviewer recommended the author describe the modifications and advantages of the present work over the others in the introduction part.
5. To show the stability of these devices, the reviewer recommended adding some stability results.
6. The author showed UV-vis absorption spectra of photosensitive layer heterojunctions under 0%, 50%, and 100% stretched states. How about the UV-vis absorption spectra after release?
7. What's the boundary between scotopic and photopic adaption? The author used the current change ratio (CCR) to distinguish the two concepts. Is there an explanation for that in biological systems?

8. In Figure 2e, the photocurrent decreases from the dim light to the bright light. This phenomenon is seldom seen from other research, what is the reason for that?

9. In supplementary Figure 19, when the V_g is at 0V, the photocurrent also shows a decrease. What is the reason for it?

10. The above phenomenon is very interesting. What is the possible cause for two different methods (voltage and light intensity) could get similar results?

11. The author only showed the comparison between the neat and hybrid perovskite layer-based devices.

To fully understand the working mechanism of the device, the reviewers also recommend adding the device performance comparison among CSPbBr₃ QCM /DPP-DTT CONPHINE film, DPP-DTT CONPHINE film, and CSPbBr₃ QCM-based devices.

12. In the mechanism part, the author said, "With the increase of illumination intensity, the photogenerated excitons were liberally produced, in which the photogenerated electrons gradually filled the defect states and the excess electrons tended to recombine with photogenerated holes (Fig. 3i)".

Then is it possible that when the photogenerated electrons fill the defect state, the newly generated photogenerated electron-hole pairs will recombine after separation? Does photocurrent stay saturated in the channel?

If so, it is hard to explain the drop in photocurrent.

Therefore, the reviewer recommended that the author explain more about the mechanism.

13. Also, the work is similar to the other work, NATURE COMMUNICATIONS | (2021) 12:1798, which also uses two layers (CsPbBr₃+CNT) as the channel. However, in the paper, there is no decrease in photocurrent. Then what is the possible reason for that?

14. In the paper, the author mentioned the main mechanism is based on the defect-tunable heterojunction. To make it more clear to the reader, the reviewer recommended that the author explain more about the defect-tunable heterojunction.

Reviewer #2 (Remarks to the Author):

The manuscript presents a notable advancement in the field of neuromorphic vision systems by introducing an intrinsically stretchable neuromorphic vision-adaptive transistor (ISNVaT) using viscoelastic perovskite films. The work is pioneering in its approach to combine strain insensitivity with quasicontinuous microsphere (QCM) morphologies, which holds promise for future applications in various fields. However, there are several major issues that need to be addressed to support the manuscript's claims effectively and to increase the reproducibility and reliability of the results presented.

1. Rationale for Material Choice and Fabrication Process:

The paper would be significantly strengthened by providing a clear justification for the selection of perovskite materials in the ISNVaT. Additionally, a detailed description of the fabrication process of the viscoelastic perovskite films, including any unique environmental or processing conditions, would enhance the reproducibility of the work.

2. Statistical Validation:

The number of samples tested is not mentioned, which is crucial for assessing the statistical significance of the results. Providing information on the sample size, along with the statistical methods used to analyze the data, will lend credibility to the findings and ensure the study's robustness.

3. Figure Quality:

The resolution and legibility of the figures included in the manuscript are inadequate for a thorough evaluation of the results. The authors should ensure high-quality, readable figures even when zoomed out to facilitate the understanding of the readers.

4. Clarity and Consistency in Metrics:

There appears to be a deviation from established formulas, specifically in the calculation of the current change ratio (CCR). A detailed explanation for the chosen formula, including the reasoning behind a 10-second illumination period, is warranted to uphold the scientific rigor of the study.

5. Illumination Conditions:

The fixed illumination value of 356 μW used in the experiments should be justified with respect to its relevance to real-world conditions or comparison to other studies in the field. Clarification on this parameter is vital for ensuring that other researchers can accurately replicate the study.

6. Diversity of Test Scenarios:

The usage of a single pattern (basketball) as a test stimulus is limited. Including a diverse array of patterns and stimuli will demonstrate the versatility of the device and solidify the results' generalizability.

Reviewer #3 (Remarks to the Author):

The authors report an intrinsically stretchable neuromorphic vision-adaptive transistor. A mixture of SEBS and perovskite quantum dots is used to form viscoelastic perovskite film with a strain-insensitive quasi-continuous microsphere morphology. This viscoelastic perovskite film is used as a photosensitive layer in stretchable neuromorphic vision-adaptive transistor due to its intrinsic stretchability, photosensitivity and tunable charge-trapping defects that enable photoadaptation and synaptic properties. The fabricated transistor exhibits a trichromatic photoadaptation, high stretchability, low energy consumption and high paired-pulse facilitation index. In addition, vision-adaptive function of human eyes is demonstrated using the neuromorphic transistor that shows a faster adaptation speed than human eyes. It is impressive that the stretchable neuromorphic transistor shows synaptic properties with low energy consumption. This is because energy efficiency is one of the important factors in autonomous applications such as visual prosthetic, robotics and unmanned vehicles. However, at the same time, it is also true that novel aspects were not highlighted much. In addition, there is insufficient data to prove the authors' claims. Therefore, the reviewer cannot be supportive for the publication of this work in Nature communications, at least in a current form. Some detailed comments that may be helpful to improve the quality of this work are as follows.

Comments #1: The primary emphasis of this paper lies in the development of an intrinsically stretchable neuromorphic optoelectronic device. To strengthen this aspect, the authors should discuss deeper into why intrinsic stretchability is pivotal.

Comments #2: The authors claim that the perovskite film can provide tunable charge-trapping defects. However, there is no data supporting the defect-tunability of the perovskite film. AFM and optical measurement data presented by the authors cannot prove the defect-tunability. It is recommended to add quantitative data that can prove the defect density of the film, such as hole-only devices measurement and Nyquist plots.

Comments #3: There are no data showing the viscoelastic properties of the perovskite film. It is recommended to add stress-strain curve and modulus data for the perovskite film.

Comments #4: The authors should further explain the mechanism of photoadaptation in more detail for better understanding with appropriate data. Why do wavelengths as well as gate voltage and light intensity affect the synaptic properties of the device? Figs. 3g-i do not match the description in the text.

Comments #5: While the authors noted a fast adaptation speed of 150 seconds, a more comprehensive comparison could be made by benchmarking it against existing devices capable of photopic and scotopic adaptation (e.g., Nat Electron 5, 84-91 (2022) etc), rather than merely contrasting it with the natural human eye's adaptation speed.

Comments #6: Furthermore, the claim that the device is strain-insensitive should be clarified, as the adaptation behaviors, characterized by CCR, are influenced by strain-induced changes. The authors need to provide more details in this regard.

Comments #7: In the manuscript, the author showed that CCR varies not only with light intensity but also with gate voltage. How was this gate voltage dependency utilized during the vision-adaptation behaviors demonstration? Please provide more details on the vision-adaptation behaviors demonstration.

Comments #8: The authors mentioned the trichromatic neuromorphic vision adaptation sensor in the paper. However, there is limited data regarding the wavelengths. While utilizing a trichromatic vision system aims to mimic the human eye, the demonstration used only UV light instead of R, G, B lights. It would be beneficial to include more details implemented under R, G, B lights in the discussion. Moreover, in Supplementary Movie 4, the significance of turning on LEDs of different colors seems ambiguous. Shouldn't the color of the input light be varied instead of changing the color of the output LED?

Responses (R) to the Comments (C)

Responses to the reviewer 1

C: *In this manuscript, the authors used a defect-tunable viscoelastic perovskite film to fabricate intrinsically stretchable neuromorphic vision-adaptive transistors. The reviewer believes that researchers in this field will be interested in the contents of this manuscript. However, the mechanism part is not very clear to the reader. Thus, the reviewer suggests that this manuscript is suitable for publication in Nature Communication after minor revision. To strengthen the manuscript, several points seemed to be discussed in detail.*

R: We sincerely appreciate the reviewer's comments and giving us the opportunity to further improve our manuscript. We carefully revised the manuscript and addressed all the concerns of the reviewer in the revised manuscript. The point-to-point answers for reviewer's comments are included in the following parts.

C1: *In this paper, the author used the transfer method to fabricate the devices. What is the yield? The reviewer recommended adding some statistical data.*

R1: We are grateful for the reviewer's constructive comments and suggestions. Benefiting from the viscoelasticity of optimized perovskite dot films, we could transfer the photosensitive layer with near 100% success rate. In fact, we totally fabricated 65 devices from 5 batches with a yield of 92.3%. In corresponding to the reviewer's suggestion, we have attached the relevant description and data in the supplementary information (Methods, Page 2, line 24).

C2: *What are the thickness for each layer?*

R2: We sincerely appreciate the reviewer's careful reading and useful suggestions. Our devices are composed of six intrinsically stretchable active layers, which are the viscoelastic perovskite photosensitive layer, CNT source/drain electrodes, hybrid semiconductor layer, hybrid dielectric layer, CNTs gate electrode, and SEBS substrate. The measured thickness of each layer was about 30 nm, 50 nm, 50 nm, 1800 nm, 80 nm and 2 mm, respectively. The relevant description and data have been added in the Supplementary Information (Methods, Page 2, line 19-22).

C3: *There is only 0% and 100% result for OM data with stretchability. To show the stretchability more obviously, the reviewer recommended adding the intermediate state, like 20%, 30%, 50%, 70%, etc...*

R3: We are grateful for the valuable comments. In response to the reviewer's suggestion, we have added the optical microscope images of the intermediate state as follows:

Figure R1. Optical microscope images of the neat PVD films and PVD/SEBS hybrid films under different strain.

As shown in figure R1, the neat film showed small cracks at 25% strain and occurred rapid crack propagation at 100% strain, while the hybrid film consistently maintained excellent stretchability without obvious cracks under various stains. These results further proved that the introduction of SEBS elastomer could enhance the viscoelasticity and stretchability of hybrid perovskite films. As the reviewer suggested, the optical microscope images have been added in Supplementary Information (Fig. S6).

C4: *The method that perovskite materials combine with SEBS to improve the stretchability has been reported by other paper, Adv. Mater. 2020, 32, 2001989. Therefore, this method is not new. The reviewer recommended the author describe the modifications and advantages of the present work over the others in the introduction part.*

R4: We thank the reviewer for pointing out this issue, which is crucial significance to improve our manuscript. Different from the reported stretchable color-conversion layer in the reference (*Adv. Mater. 2020, 32, 2001989.*), we mainly focused on the retained photosensitive and defect-tunable properties enabled by the novel quasi-continuous microsphere (QCM) morphology for multifunctional neuromorphic visual devices. According to the reviewer's suggestion, we have added the corresponding explanations and cited the literature in the revised manuscript (Page 3, Line 1-3).

C5: *To show the stability of these devices, the reviewer recommended adding some stability results.*

R5: We sincerely appreciate the reviewer's valuable suggestions. In accordance with the reviewer's suggestion, we have measured the operational and storage stability as follows:

Figure R2. Stability test of ISNVaTs. (a) Cycle test stability at dark; (b) Cycle test stability under 365 nm illumination ($P_{in}=538.85 \mu\text{W cm}^{-2}$); (c) Temporal stability in nitrogen atmosphere for 14 days ($P_{in}=538.85 \mu\text{W cm}^{-2}$).

The results indicated that our device could maintain excellent photoelectrical performance even after the repeated operational cycles and long-term storage. As the reviewer suggested, we have attached the corresponding stability explanations and figures in the revised manuscript (Page 6, Line 28-29) and Supporting Information (Fig. S16).

C6: The author showed UV-vis absorption spectra of photosensitive layer heterojunctions under 0%, 50%, and 100% stretched states. How about the UV-vis absorption spectra after release?

R6: We appreciate the reviewer's helpful comments. In response to the reviewer's suggestion, we have further characterized the UV-vis absorption spectra of the photosensitive layer heterojunctions under 0%, 50%, and 100% strain and after releasing the strain as follows:

Figure R3. The photosensitive layer heterojunction under 0%, 50%, 100% strains and after the release.

As shown in figure R3, the photosensitive layer heterojunctions behaved ideal absorption even under 100% strain and release states that enabled relatively high-effective photoelectric conversion and charge transport for our stretchable phototransistors. In accordance with the suggestions of the reviewer, we have modified the relevant figure and description in the revised manuscript (Page 6, Line 9) and

Supplementary Information (Fig S10 b).

C7: What's the boundary between scotopic and photopic adaption? The author used the current change ratio (CCR) to distinguish the two concepts. Is there an explanation for that in biological systems?

R7: We thank the reviewer for pointing out the distinction between scotopic and photopic adaption concepts, which is significant to improve the quality of our manuscript. In fact, there is no specific range of the scotopic and photopic adaption for biological systems due to the considerable difference and complicated modulation in various environment and configurations. Analogizing with biological adaptive systems, we categorized the adaptation into several processes as shown in Figure R4: I. Reception and perception, II. Transformation, III. Adaptation (*Book: Information Processing: Retinal Adaptation*). After the reception and perception of the external environment, adaptive behaviors would occur in the biological systems by transforming specific cellular functions. Typically, the transition point from cone-mediated to rod-mediated thresholds was termed the rod-cone break, which was almost about one fifth (10 min) of the entire adaptation time (45-50 min).

Figure R4. Schematic diagram of the adaptation process. (a) Scotopic adaptation. (b) Photopic adaptation. (I. Reception and perception; II. Transformation; III. Adaptation.)

Therefore, to fairly evaluate the adaption behaviors of bionic devices, the current change ratio (CCR) was defined at the rod–cone break time in our manuscript, while it was defined at the initial state in the reported references (*Nat. Electron. 2022, 5, 84–91*). Additionally, we considered it was a relative comparison of scotopic and photopic adaptation to better describe the real situation of visual adaptation at a benchmark condition (CCR=1). Referring to these mentioned references, we employed the definition of CCR and amended it to fit the biologic properties more closely in our manuscript. To make it more clear to the readers, we have added the relevant descriptions and figures in the revised manuscript (Page 8, Line 9) and Supplementary Information (Fig. S23).

C8: In Figure 2e, the photocurrent decreases from the dim light to the bright light. This phenomenon is seldom seen from other research, what is the reason for that?

R8: We apologize for the confusion about the photocurrent and appreciate the reviewer's careful reading. In fact, the absolute photocurrent value of our device in bright light is higher than the value in the dim light as shown in Figure R5. However, to intuitively compare the current adaptive trend under different ambient lights, we utilize the normalized current in Figure 2e, respectively. To avoid any misunderstanding to readers, we have modified the figure and captions of absolute photocurrent value in our manuscript (revised Figure 2e) and added the single-curve normalized current figures in Supplementary Information (Fig. S22).

Figure R5. Light intensity dependence (top, $V_G=0$) and gate voltage dependence (bottom, $P_{in}=356.69 \mu\text{W cm}^{-2}$) adaptation behaviors of ISNVaTs (dim: $6.41 \mu\text{W cm}^{-2}$, normal: $356.69 \mu\text{W cm}^{-2}$ and bright: $698.09 \mu\text{W cm}^{-2}$; positive: $+10\text{V}$, normal: 0V and negative: -10V).

C9: *In supplementary Figure 19, when the V_g is at 0V , the photocurrent also shows a decrease. What is the reason for it?*

R9: We are grateful for the reviewer to point out the photocurrent variation issue. Theoretically, we could obtain a constant photocurrent value at $V_G = 0\text{V}$ at an environmental light condition where the CCR is equal to 1. However, the measured photocurrent has a small decreasing trend that could be caused by the slight deviation of light intensity from the illumination equipment. As a result, the calculated CCR value was closed to 1 (CCR=0.967 as shown in the revised Fig. S25) at $V_G = 0\text{V}$. Therefore, the slightly decreased photocurrent would not affect the reliability of the gate voltage dependence.

C10: *The above phenomenon is very interesting. What is the possible cause for two different methods (voltage and light intensity) could get similar results?*

R10: We appreciate the reviewer's valuable comments to help us improve our manuscript. Typically, the gate voltage dependence behaviors benefit from the self-modulation of quasi-Fermi levels, while the light intensity dependence behaviors originate from the dynamic change of photogenerated carrier concentration (*Nat. Electron.* 2021, 4, 522–529; *Adv. Mater.* 2022, 34, 2205679.). Although their mechanisms are different, the direct modulation on the photocurrent variation are both achieved by tuning the carrier (or defect state) density in the channel (*Nat Electron* 2022, 5, 84-91). Specially, the photogating effect (a conductance modulation through

photoinduced gate voltage), which could be ascribed to the prolonged excess carrier lifetime induced by defects and impurities or artificial designed hybrid structures, has been reported in other researches (*Adv. Sci.* 2017, 4, 1700323.; *Nat. Commun.* 2024, 15, 141.). Therefore, the final adaption results are similar in the two different routes.

C11: *The author only showed the comparison between the neat and hybrid perovskite layer-based devices.*

To fully understand the working mechanism of the device, the reviewers also recommend adding the device performance comparison among CsPbBr₃ QCM /DPP-DTT CONPHINE film, DPP-DTT CONPHINE film, and CsPbBr₃ QCM-based devices.

R11: We fully appreciate the reviewer's valuable suggestions to compare experimental results with different active layers. The device performance comparison among QCM QD films, DPP-DTT CONPHINE films, and QCM QD films/DPP-DTT CONPHINE films has been added in Figure R6 as recommended.

Figure R6. Comparison of typical transfer curves on various active layers. (a) QCM QD films; (b) DPP-DTT CONPHINE films; (c) QCM QD films/DPP-DTT CONPHINE films.

Typically, there was almost no transistor characteristic based on the QCM QD films, while the DPP-DTT CONPHINE films displayed ideal transfer curves and had unobvious photoresponse under 365 nm illumination ($P_{in}=538.85 \mu\text{W cm}^{-2}$). However, the novel elastic photosensitive heterojunctions based on QCM QD films and DPP-DTT CONPHINE films demonstrated excellent photoresponse, which benefited subsequent optical neuroelectronic applications. In this revised manuscript, we have added the relevant figures in the Supplementary Information (Fig. S18).

C12: *In the mechanism part, the author said, “With the increase of illumination intensity, the photogenerated excitons were liberally produced, in which the photogenerated electrons gradually filled the defect states and the excess electrons tended to recombine with photogenerated holes (Fig. 3i)”*

Then is it possible that when the photogenerated electrons fill the defect state, the newly generated photogenerated electron-hole pairs will recombine after separation? Does photocurrent stay saturated in the channel?

If so, it is hard to explain the drop in photocurrent.

Therefore, the reviewer recommended that the author explain more about the mechanism.

R12: We appreciate the careful reading and aculeate comments from the reviewer. Indeed, the recombination of excitons always exists after the separation of photogenerated electron-hole pairs. As shown in Figure R7, under constant bright light illumination, a saturation phenomenon initially occurs with the dominance of photogenerated carriers (Region I), and then the defect states (or exciton recombination) play a key role in the subsequent process (Region II). When the photogenerated electrons fill the defect states or recombine with photogenerated holes, the saturated current gradually decrease. However, the recombination rate and charge numbers are limited by the intrinsic trap states and the heterojunction effect. Therefore, the decreased current also will become saturated (Region III). To avoid any misunderstanding, we have added the relevant explanations in this revised manuscript (Page 12, Line 11-15).

Figure R7. Mechanistic diagram of photopic adaptation in three specific regions.

C13: Also, the work is similar to the other work, *NATURE COMMUNICATIONS* | (2021) 12:1798, which also uses two layers ($\text{CsPbBr}_3 + \text{CNT}$) as the channel. However, in the paper, there is no decrease in photocurrent. Then what is the possible reason for that?

R13: We sincerely appreciate the reviewer's valuable comments. The mentioned work combined CNTs and CsPbBr_3 -QDs as active heterojunction for a flexible optoelectronic sensor array to achieve optical neuromorphic functions. However, the excessive and uncontrollable defect states could be the reason for preventing the device from adaptive behaviors. In contrast, the hybrid elastic perovskite films with QCM morphology could provide tunable defect states to construct diverse neuromorphic functions. Moreover, the novel elastic photosensitive heterojunctions enable ideal heterointerface for highly effective carrier transport. Therefore, the ISNVaTs demonstrated both excellent scotopic and photopic adaptation rather than a single phenomenon of optical gain.

C14: In the paper, the author mentioned the main mechanism is based on the defect-tunable heterojunction. To make it more clear to the reader, the reviewer recommended that the author explain more about the defect-tunable heterojunction.

R14: We are grateful for the reviewer’s meaningful suggestions to improve our manuscript. Typically, efficient charge transport of heterojunctions is favored by ideal interfacial contact, matchable energy levels, and intimate morphology among heterogeneous materials with fewer defect states. However, endowing photosensitive heterojunctions with tunable defects is still challenging for neuromorphic electronics. In this work, we constructed a novel elastic photosensitive heterojunction based on nanoconfined polymer and QCM perovskite films with tunable morphologies, where the heterointerface enabled highly effective carrier transport and the optimized morphology provided tunable defect states. The defect modulation of the photosensitive heterojunction could combine high photosensitivity with neural functions, while the tunable defect states and Fermi energy levels resulting from heterojunction contacts rationally modulate the carrier transport process in functional devices (Fig. R8). From the transient photoluminescence decay spectra (Fig. 1g), the defect state tunability was well proved by the evolved decay lifetime. Additionally, the hole-only devices measurement was also applied to quantify the defect state density (N_t) of different perovskite films (Supplementary Fig. 11 and Supplementary Table 3). To clarify this issue, we have added the relevant explanations in the revised manuscript (Page 6, Line 12-17) and Supplementary Information (Fig.S35).

Figure R8. Mechanism diagram of defect-tunable heterojunction.

Responses to the reviewer 2

C: *The manuscript presents a notable advancement in the field of neuromorphic vision systems by introducing an intrinsically stretchable neuromorphic vision-adaptive transistor (ISNVaT) using viscoelastic perovskite films. The work is pioneering in its approach to combine strain insensitivity with quasi-continuous microsphere (QCM) morphologies, which holds promise for future applications in various fields. However, there are several major issues that need to be addressed to support the manuscript's claims effectively and to increase the reproducibility and reliability of the results presented.*

R: We sincerely appreciate the reviewer's positive and insightful comments, which are of crucial significance to our manuscript. We have carefully revised the manuscript and addressed all the issues raised by the reviewers. Thanks again for your careful reading, valuable comments and helpful suggestions.

C1: *Rationale for Material Choice and Fabrication Process: The paper would be significantly strengthened by providing a clear justification for the selection of perovskite materials in the ISNVaT. Additionally, a detailed description of the fabrication process of the viscoelastic perovskite films, including any unique environmental or processing conditions, would enhance the reproducibility of the work.*

R1: We sincerely appreciate the reviewer's carefully reading, valuable comments and constructive suggestions. The selected CsPbBr₃ perovskite QDs are commercially available materials (Nanjing MKNANO Tech. Co. Ltd.), which provide sufficient theoretical basis and convenience for our research. Additionally, the energy levels of the CsPbBr₃ QDs (LUMO=-3.45 eV, HOMO=-5.82 eV) are well-matched with our stretchable polymer semiconductors DPP-DTT (LUMO= -3.9 eV, HOMO= -5.1 eV), enabling high-efficiency separation of photogenerated excitons and thus achieving outstanding photosensitivity. Moreover, CsPbBr₃ QDs could be well dispersed in non-polar solvents and have good miscibility with SEBS elastomer in the toluene solvent, facilitating the subsequent morphology modulation. Therefore, the excellent photovoltaic conversion efficiency, matched energy levels, and good compatibility with elastomers collectively enable the CsPbBr₃ perovskite QDs quite desirable for use in photosensitive layers. (Methods, Page 1, Line 31-33)

As for the detailed description of the viscoelastic perovskite films fabrication process as follows, we have added in Supplementary Information (Methods, Page 1, Line 28-41).

Preparation of hybrid photosensitive films

The SEBS-H1221 was directly added to the purchased CsPbBr₃ QDs solution (10 mg/mL, n-hexane) with different weight ratios (SEBS-H1221: CsPbBr₃ QDs) of 10:1, 10:2 and 10:3, followed by continuous stirring at room temperature (around 25 °C) for 1 h. The excellent photovoltaic conversion efficiency, matched energy levels, and good compatibility with elastomers collectively enable the CsPbBr₃ perovskite QDs quite desirable for use in photosensitive layers. To tune the morphology and microphase separation of hybrid films, a high-qualified surface-hydroxylation

PDMS (SHPDMS) substrate was essential, which was fabricated by spin-coating PDMS (base/cross-linker = 10:1, w/w) solution (diluted in cyclohexane for 0.5 mg/mL) onto the Si wafer. Then modulating the surface energy of SHPDMS with different treatment times of O₂ plasma on the surface of PDMS substrates, the hybrid PQD solution was spin-coated on the substrate at 1500 rpm for 40s without annealing treatment. As a result, the QCM perovskite films were successfully fabricated on the substrate with O₂ plasma treatment around 6 min (30W, 0.5 air pressure).

C2: Statistical Validation: *The number of samples tested is not mentioned, which is crucial for assessing the statistical significance of the results. Providing information on the sample size, along with the statistical methods used to analyze the data, will lend credibility to the findings and ensure the study's robustness.*

R2: We fully understand the reviewer's concern about the statistical validation. In order to address this issue, we have added multi-sample data for light intensity dependence and gate voltage dependence in Figure R9.

Figure R9. Statistical data pots of adaptation behavior. (a) Light intensity dependence that CCR at different P_{in} values. (b) Gate voltage dependence that CCR at different V_G values. (N=6)

As shown in Figure R9, our devices exhibited relatively obvious adaptation and reliable light-intensity/gate-voltage tunable behaviors with allowable errors, demonstrating the potential of ISNVaTs for next-generation wearable neuromorphic equipment. As a result, we have added the relative explanations and figures in the revised manuscript (Page 8 Line 27) and Supplementary Information (Fig. S26).

C3: Figure Quality: *The resolution and legibility of the figures included in the manuscript are inadequate for a thorough evaluation of the results. The authors should ensure high-quality, readable figures even when zoomed out to facilitate the understanding of the readers.*

R3: We appreciate the reviewer for pointing out this issue to help our manuscript to be better understood. We have reviewed the manuscript with related images and provided high-resolution figures in the revised manuscript for better clarity, readability and comprehension.

C4: Clarity and Consistency in Metrics: *There appears to be a deviation from established formulas, specifically in the calculation of the current change ratio (CCR). A detailed explanation for the chosen formula, including the reasoning behind a 10-second illumination period, is warranted to uphold the scientific rigor of the study.*

R4: We sincerely thank the reviewer’s valuable comments. We fully agree with the reviewer’s demands for scientific rigor and believe that more detailed explanations would improve the quality of our manuscript. In fact, the visual adaptation would both go through a complete process of external light reception, light intensity perception, cell transformation and adaptation, respectively. Referring to the book (*Information Processing: Retinal Adaptation*), taking scotopic adaptation as an example, the transition point from cone-mediated to rod-mediated thresholds was termed as the rod–cone break, and it was almost about one fifth (10 min) of the entire adaptation time (45-50 min). To clarify the issue, we provided a simple schematic diagram as shown below (Figure R10).

Figure R10. Schematic diagram of the adaptation process. (a) Scotopic adaptation. (b) Photopic adaptation. (I. Reception and perception; II. Transformation; III. Adaptation.)

Analogizing with biological adaptive systems, we categorized the adaptation into several processes: I. Reception and perception, II. Transformation, III. Adaptation. It can be noticed that the rod–cone break typically occurs near 10 s (the end of process II) Therefore, we defined CCR based on the photocurrent at 10 s rather than the initial current (*Nat Electron 2022, 5, 84-91*), which was believed to be more in consistent with the real situation time of visual adaptation. We have added the relevant descriptions, figures and captions in Supplementary Information (Fig. S23) for a clearer explanation.

C5: Illumination Conditions: *The fixed illumination value of 356 μW used in the experiments should be justified with respect to its relevance to real-world conditions or comparison to other studies in the field. Clarification on this parameter is vital for ensuring that other researchers can accurately replicate the study.*

R5: We are grateful for the reviewer’s constructive suggestions on the illumination conditions, which are crucial for improving our manuscript. Lux (lx) is a commonly used unit for illumination in the real environment. According to the measurement and analysis in the reported reference (*Journal of Measurements in Engineering. 2020. 8(3)*), the irradiance to illuminance conversion factor is 120 lx equals 1 W m^{-2} . Therefore, the illumination value of $356 \mu\text{W cm}^{-2}$ used in our manuscript can be converted to 427.2 lux, which is consistent with the comfortable indoor illumination on sunny days (100-1000 lux) (*Book: Advances in Design for Inclusion (pp.180-191)*). This directly demonstrates the practical significance of the fixed illumination value used in real-

world scenarios. The relevant descriptions have been added in the Supplementary Information (caption of Fig.S25).

C6: *Diversity of Test Scenarios: The usage of a single pattern (basketball) as a test stimulus is limited. Including a diverse array of patterns and stimuli will demonstrate the versatility of the device and solidify the results' generalizability*

R6: We appreciate the valuable recommendations of the reviewer. In response to the reviewer's suggestion, we additionally imaged the other patterns (including the sun and house shapes) with our ISNVaTs under green (520 nm) and blue light (460 nm), respectively (Figure R11).

Figure R11. Single-pixel adaptive imaging of ISNVaTs. (a) Simulated visual adaptation results under green light with a house-shaped mask; (b) Simulated visual adaptation results under blue light with a sun-shaped mask. (illumination intervals: 10 ms, 100 ms and 1 s)

As shown in Figure R11, the ISNVaTs demonstrated excellent adaptation behaviors with multiple pattern images under various light conditions. Consequently, those results have proved the reliability and generalizability of the single-pixel adaptive imaging of our device. In this revised manuscript, we have modified the relevant explanations and added the figures in the Supplementary Information (Fig. S40).

Responses to the reviewer 3

C: The authors report an intrinsically stretchable neuromorphic vision-adaptive transistor. A mixture of SEBS and perovskite quantum dots is used to form viscoelastic perovskite film with a strain-insensitive quasi-continuous microsphere morphology. This viscoelastic perovskite film is used as a photosensitive layer in stretchable neuromorphic vision-adaptive transistor due to its intrinsic stretchability, photosensitivity and tunable charge-trapping defects that enable photoadaptation and synaptic properties. The fabricated transistor exhibits a trichromatic photoadaptation, high stretchability, low energy consumption and high paired-pulse facilitation index. In addition, vision-adaptive function of human eyes is demonstrated using the neuromorphic transistor that shows a faster adaptation speed than human eyes. It is impressive that the stretchable neuromorphic transistor shows synaptic properties with low energy consumption. This is because energy efficiency is one of the important factors in autonomous applications such as visual prosthetic, robotics and unmanned vehicles. However, at the same time, it is also true that novel aspects were not highlighted much. In addition, there is insufficient data to prove the authors' claims. Therefore, the reviewer cannot be supportive for the publication of this work in *Nature communications*, at least in a current form. Some detailed comments that may be helpful to improve the quality of this work are as follows.

R: We sincerely appreciate the reviewer's valuable and insightful comments, which are of crucial significance to our manuscript. We have gratefully received the reviewer's suggestions and carefully addressed the concerns in the revised version. Thanks again for the review's careful reading, valuable comments, and helpful suggestions, and we are looking forward to a positive response from the reviewer.

C1: The primary emphasis of this paper lies in the development of an intrinsically stretchable neuromorphic optoelectronic device. To strengthen this aspect, the authors should discuss deeper into why intrinsic stretchability is pivotal.

R1: We appreciate the reviewer for pointing out this issue to help our manuscript to be better understood. In fact, it is acknowledged that intrinsic stretchability can provide high elastic deformation (stretching, twisting, poking, etc.) and cross-scale modulus adaptability. As typically achieved by the molecular-level deformation, intrinsically stretchable materials and devices are free of the complicated geometric engineering, and thus resulting in convenient fabrication, versatility, and high availability. Moreover, intrinsically stretchable electronics have demonstrated excellent portability, biocompatibility and interface adhesiveness, which will lead to exciting advances and opportunities for broad applications. Notably, intrinsically stretchable neuromorphic optoelectronics, as an inevitable component for the next-generation smart wearables, hardware implementations and humanoid robotics, will undoubtedly be extremely significant for future human lifestyles. As the reviewer suggested, we have added the corresponding explanations in the revised manuscript (Page 2 line 16-20).

C2: The authors claim that the perovskite film can provide tunable charge-trapping defects. However, there is no data supporting the defect-tunability of the perovskite film.

AFM and optical measurement data presented by the authors cannot prove the defect-tunability. It is recommended to add quantitative data that can prove the defect density of the film, such as hole-only devices measurement and Nyquist plots.

R2: We appreciate the reviewer for constructive comments and pointing out the feasible measurement to certify the defect-tunability of the perovskite films. Following the reviewer's suggestion, we have applied hole-only device measurement by constructing a device with the Ag/MoO₃/Viscoelastic PQD films/PEDOT:PSS/ITO/Glass configuration (Figure R12).

Figure R12. Defect-tunability of different viscoelastic perovskite films. (a) Schematic diagram of device structure for the hole-only devices measurement; (b) SCLC curves of different viscoelastic perovskite films.

The defect state density is calculated as follows:

$$V_{TFL} = \frac{eN_t L^2}{2\epsilon\epsilon_0}$$

where e is the elementary charge, L is the thickness of the semiconductor film, ϵ_0 is the vacuum dielectric constant, and ϵ is the relative dielectric constant (*Adv. Sci.* 2022, 9, 2105856). The fetched V_{TFL} and thus calculated N_t of various viscoelastic perovskite films are concluded in the Table R1.

Table R1. The V_{TFL} and N_t among various viscoelastic perovskite films.

	V_{TFL} (V)	N_t (cm ⁻³)
Neat	1.71	5.62×10^{18}
10:1 Spindle-like	2.42	7.96×10^{18}
10:1 Honeycomb-like	2.07	6.81×10^{18}
10:1 QCM	1.76	5.79×10^{18}
10:2	3.03	9.96×10^{18}
10:3	3.28	1.08×10^{19}

From the extracted values, the N_t obviously increased from 5.62×10^{18} cm⁻³ for neat films to 1.08×10^{19} cm⁻³ for hybrid films (10:3), and it demonstrates tunability from the spindle-like morphology (7.96×10^{18} cm⁻³) to QCM morphology (5.79×10^{18} cm⁻³). Therefore, the defect states are certainly introduced by the elastomer and can be

effectively tuned through surface energy-induced strategy. The hole-only device measurement largely proves the reliability of our strategy to achieve defect-tunable elastic photosensitive films. The detailed calculation about the defect states and related figures have been provided in the revised manuscript (Page 6, Line 12-17) and the Supplementary Information (Fig. S11 and Table S3).

C3: *There are no data showing the viscoelastic properties of the perovskite film. It is recommended to add stress-strain curve and modulus data for the perovskite film.*

R3: We appreciate the practical recommendations provided by the reviewer for validating the viscoelastic properties of the perovskite film. In fact, we have tried to measure the stress-strain behaviors of perovskite films. However, it is difficult to be peeled off into free-standing films due to the specific preparation by surface energy modulation and the low film thickness (about 30 nm). To address the reviewer's concerns, we have provided the modulus measurement in Figure R13.

	Hight	DMT Modulus	DMT Modulus (MPa)		Hight	DMT Modulus	DMT Modulus (MPa)
Neat 0 min			267	Hybrid 0 min			69.7
Neat 3 min			686	Hybrid 3 min			191
Neat 6 min			1244	Hybrid 6 min			113

Figure R13. DMT modulus measurement of neat and hybrid perovskite films (PQDs: SEBS = 10:1, w/w) under different O₂ Plasma treatment.

It is acknowledged that the elastic modulus can also reflect the elastic deformation capability of thin films. As shown in Figure R13, the elastic modulus of perovskites films substantially reduces from 267 to 69.7 MPa, which demonstrates the improved stretchability after blending with elastomers. Furthermore, the calculation of adhesion work in Supplementary Table 2 further demonstrates improved viscoelastic properties from neat films (60.13 mJ/m²) to hybrid films (10:3, 67.99 mJ/m²), thus proving that the optimized hybrid perovskites film has a favorable surface energy for the ideal interfacial contact. Additionally, the SEBS elastomers have high intrinsic viscoelasticity so that the viscoelastic property of hybrid perovskite films with high SEBS content can be effectively improved. In the revised manuscript, we have provided the related explanations (Page 4, Line 18-23) and figures in Supplementary Information (Fig. S7).

C4: *The authors should further explain the mechanism of photoadaptation in more detail for better understanding with appropriate data. Why do wavelengths as well as gate voltage and light intensity affect the synaptic properties of the device? Figs. 3g-i*

do not match the description in the text.

R4: We are grateful for the reviewer to point out the ambiguous explanation, which could be helpful for improving our manuscript. Actually, Figs.3g-i described the mechanism of various adaptive behaviors that exhibited gate-voltage dependence and light-intensity dependence. Additionally, the synaptic behavior strongly depends on the incident light intensity, light duration and interval time due to the current relaxation. The adaptive phenomenon of ISNVaTs could be depicted as a dynamic trapping and de-trapping process of carriers, which mainly resulted from the abundant defect states.

Wavelength dependence: The photosensitive heterojunction based on CPB QDs and DPP-DTT/SEBS film was conducted to further broaden spectral absorption (CPB QDs: 300-520 nm; DPP-DTT: 600-1000 nm). However, as quantum dots and polymer semiconductors have different photoelectric conversion efficiencies, the ISNVaTs would show successively decreased efficiency from 365 nm light, 460 nm light, 520 nm light to 625 nm light. Therefore, the ISNVaTs would exhibit faster adaptive processes under ultraviolet ray than visible light, which resulted in recognizable neuromorphic properties.

Figure R14. Mechanism diagram of wavelength dependence. (a) Adaptation under UV light; (b) Adaptation under blue light; (c) Adaptation under green light.

Light intensity dependence: When a dim light is applied at the initial state ($V_G=0$), the photogenerated excitons would be separated at the interface, while the photogenerated holes would be easily injected into the conductive channel to enable more efficient charge transport and the electrons would be trapped by the defect states. With the illumination intensity increased, the defect states tend to be filled, and the trapping and de-trapping processes would be in a dynamic balance, resulting in an almost stable current. As illumination intensity continually increased, however, the defect states are immediately filled with photogenerated electrons, and the rest electrons would bring an enrichment at interface and accelerate charge recombination. As a result, an apparent decreased current is caused under high light intensity.

Figure R15. Mechanism diagram of light intensity dependence. (a) Adaptation under dim light; (b) Adaptation under normal light; (c) Adaptation under bright light.

Gate voltage dependence: The Fermi level (E_F) in heterojunction will be tuned with different bias gate voltages. When an appropriate light intensity is applied, the E_F would be raised at a positive V_G that results in more defect states. Therefore, the current gradually increased due to the reduced recombination dominated by electron trapping processes. Instead, the E_F would be lowered at a negative V_G , and the reduced defect states facilitate the de-trapping processes of photogenerated electrons, implying a decreased current.

Figure R16. Mechanism diagram of gate voltage dependence. (a) Adaptation under positive gate voltage; (b) Adaptation under zero gate voltage; (c) Adaptation under negative gate voltage.

In response to reviewer and avoid misunderstandings, we have added the detailed mechanistic diagrams and related explanations in the Supporting Information (Fig. S35-37).

C5: While the authors noted a fast adaptation speed of 150 seconds, a more comprehensive comparison could be made by benchmarking it against existing devices capable of photopic and scotopic adaptation (e.g., *Nat Electron* 5, 84-91 (2022) etc), rather than merely contrasting it with the natural human eye's adaptation speed.

R5: We sincerely appreciate the reviewer's valuable comments and constructive suggestions. According to the reviewer's suggestion, we summarized and provided the comparison of adaptation performance in details (Table R2). However, there are some obstacles to evaluate adaptive results among recent reports, for example the different definitions in adaptation time that make it hard to compare these results at the same level. Notably, the adaptive functions largely rely on the active materials, defect introduction approaches and the device structures. Additionally, intrinsic stretchable organic adaptive devices are more challenging due to the technical differences in photoelectric conversion efficiency and device construction. Therefore, the adaptive speed of our device may not be on par with the advanced rigid materials, but it is still bionic-eye compatible. However, to address the reviewer's concern, we have added the related table for the comparison of adaptive time in the revised manuscript (Page 13, Line 6-7) and the Supplementary Information (Table. S7).

Table R2. Comparison of adaptive time among recent related works.

Active layer	Stretchability	Semiconductor type	Visual adaptation	Adaptation time (s)	Reference
CsPbBr ₃ QDs/MoS ₂	Rigid	2D	Photopic adaptation	> 4	10.1002/adfm.202010655
CsFAMA	Rigid	2D	Photopic adaptation	4.8	10.1002/aisy.202000122
CsPb(Br _{1-x} I _x) ₃ /MoS ₂	Rigid	2D	Photopic adaptation	> 60	10.1021/acsnano.0c01689
MOF	Rigid	2D	Photopic adaptation Scotopic adaptation	10 10	10.1021/acs.chemmater.3c01422
MoS ₂	Rigid	2D	Photopic adaptation Scotopic adaptation	80 10	10.1038/s41928-022-00713-1
Graphene/PbS QDs/graphene	Rigid	2D	Photopic adaptation Scotopic adaptation	100s	10.1002/adma.202205679
CdSe&IGZO	Rigid	Inorganic oxide	Photopic adaptation Scotopic adaptation	10 10	10.1002/adma.201906433
MAPbI ₃ &IZO	Rigid	Inorganic oxide	Photopic adaptation Scotopic adaptation	> 40	10.1002/adma.202105485
MoO ₃ /LiTFSI	Rigid	Inorganic oxide	Photopic adaptation Scotopic adaptation	> 50	10.1016/j.nanoen.2022.107142
InP QDs/ITZO	Rigid	Inorganic oxide	Photopic adaptation Scotopic adaptation	300 300	10.1002/adfm.202305959
Two bulk heterojunctions	Rigid	Organic	Photopic adaptation	2	10.1038/s41928-021-00615-8
CsPbBr ₃ /TIPS	Rigid	Organic	Photopic adaptation Scotopic adaptation	> 300	10.1021/acсами.1c11866
CsPbBr ₃ QCM film /DPP-DTT CONPHINE film	100% Stretchable	Organic	Photopic adaptation Scotopic adaptation	< 150	This work

C6: Furthermore, the claim that the device is strain-insensitive should be clarified, as the adaptation behaviors, characterized by CCR, are influenced by strain-induced changes. The authors need to provide more details in this regard.

R6: We are grateful for the reviewer's careful reading, and apologize for the confusion caused by the ambiguous references. Here, the strain insensitivity was loosely defined for the viscoelastic perovskite films that could enable stable photoelectric performance under various strain. In fact, the photoresponse of our device was observed no obvious variation during stretching (Fig. 2i-j and Fig. S28-S29). However, in the adaptation behaviors, the strain-associated changes in resistance, leakage current and defect states in the whole device might play a significant role in the enhanced bionic neurobehaviors. To avoid any misunderstanding to reader, we have modified the expression in the revised manuscript (Page 9, Line 22).

C7: In the manuscript, the author showed that CCR varies not only with light intensity but also with gate voltage. How was this gate voltage dependency utilized during the vision-adaptation behaviors demonstration? Please provide more details on the vision-adaptation behaviors demonstration.

R7: We thank the reviewer for pointing out the issues that need to be improved in our manuscript. In fact, the fabricated devices also enabled the dynamic modulation by the gate voltage, which could be complemented with light intensity dependence for better demonstrating the vision-adaptation simulation. In response to the reviewer's suggestions, we have added the current mapping of the 5*5 ISNVaT array under different gate voltages in Figure R17.

Figure R17. Visual adaptation of the 5*5 ISNVaT array under different gate voltage and under 50% strain. (a) Scotopic adaptation ($V_G = +10$ V, $P_{in} = 356.69 \mu\text{W cm}^{-2}$); (b) Photonic adaptation ($V_G = -10$ V, $P_{in} = 356.69 \mu\text{W cm}^{-2}$). (The device at the center is applied at $V_G = -10$ V as a comparison.)

As shown in Figure R17, the 5*5 ISNVaT array demonstrated excellent gate-voltage-dependence images. When a positive gate voltage was applied, the photocurrent would gradually increase as the time increased and resulted in a high contrast compared with the center-contrast device. Instead, at a negative gate voltage, the photocurrent rapidly decreased. In this revised manuscript, we have modified the relevant explanations (Page 13, Line 2-4) and added the figure in the Supplementary Information (Fig. S39).

C8: *The authors mentioned the trichromatic neuromorphic vision adaptation sensor in the paper. However, there is limited data regarding the wavelengths. While utilizing a trichromatic vision system aims to mimic the human eye, the demonstration used only UV light instead of R, G, B lights. It would be beneficial to include more details implemented under R, G, B lights in the discussion. Moreover, in Supplementary Movie 4, the significance of turning on LEDs of different colors seems ambiguous. Shouldn't the color of the input light be varied instead of changing the color of the output LED?*

R8: We are very grateful to the reviewers for pointing out the issues about the adaption demonstration. We have constructed the elastic photosensitive heterojunction with the QCM CsPbBr₃ film and DPP-DTT CONPHINE film. As shown in Figure 1f (UV-vis absorption spectra) and Supplementary Figure 19 (typical transfer curves under various light wavelengths in revised version), the ISNVaT mainly exhibited strong absorption of ultraviolet light, blue and green light due to the intrinsic photosensitive properties of the materials. Therefore, we have achieved trichromatic neuromorphic vision adaptation behaviors under aforementioned light wavelengths. (Supplementary Figure 27 in revised version). Although the device also shows a certain photoresponse to red light and infrared light, the weak photoelectric conversion efficiency of polymer semiconductor makes it insufficient to achieve neuromorphic vision adaptation simulation under these two wavelengths. Moreover, the limited reading range of measurement system also cannot support the effective images. However, in response to the reviewer's suggestions, we have additionally provided the adaptive imaging under blue and green light in Figure R18. The relevant explanations (Page 15, Line 4-6) and

figures have been added in the Supplementary Information (Fig. S40).

Figure R18. Single-pixel adaptive imaging of ISNVaTs. (a) Simulated visual adaptation results under green light with a house-shaped mask; (b) Simulated visual adaptation results under blue light with a sun-shaped mask. (illumination intervals: 10 ms, 100 ms and 1 s)

The similar results were also observed in the LED driver based on the ISNVaT pixel circuit. The implementation of the entire pixel circuit was usually accompanied by certain losses of photoresponse. Therefore, the ultraviolet light was more suitable for the LED driving. In addition, we apologized for our ambiguous expression that might have caused the reviewer's misunderstanding. The demonstration that turning on LEDs of different colors aimed to visualize the adaptive behaviors, and showed the generalizability and feasibility of ISNVaT pixels which could drive LEDs from light to dark under UV light. To address the issue raised by the reviewer, we have altered blue and green light as the input light to demonstrate more adaptive behaviors by driving LEDs (video screenshots in Figure R19). However, due to the similar reasons as mentioned, the red light is unable to achieve a favorable adaptive demonstration. In this revised manuscript, we have added the relevant description (Page 15, Line 11-12), figures and movies in the Supplementary Information (Movie S5).

Figure R19. Video screenshots of visual adaptation using an ISNVaT pixel as an LED driver. (a) Scotopic adaptation under blue light. (b) Photopic adaptation under blue light. (c) Scotopic adaptation under green light. (d) Photopic adaptation under green light.

REVIEWERS' COMMENTS

Reviewer #1 (Remarks to the Author):

In this manuscript, the authors used a defect-tunable viscoelastic perovskite film to fabricate intrinsically stretchable neuromorphic vision-adaptive transistors. The reviewer believes that researchers in this field will be interested in the contents of this manuscript. And the reviewer appreciates summarizing and comparing previous related work here. Thus, the reviewer suggests this manuscript is strengthened after the revision and is suitable for publication in Nature Communication.

Reviewer #2 (Remarks to the Author):

Most comments have been addressed and the paper can be recommended for publication.

Reviewer #3 (Remarks to the Author):

The authors have addressed the pointed comments well and the revised manuscript is now ready for publication.